# Children Learning About Secondhand Smoke (CLASS III): a protocol for a cluster randomised controlled trial of a school-based smoke-free intervention in Bangladesh and Pakistan

Rumana Huque ![ORCID] ,[1,2] Kamran Siddiqi ![ORCID] ,[3,4] Mariam Khokhar,[3] Cath Jackson,[5] Mona Kanaan,[3] Catherine Hewitt,[3] Ian Kellar,[6] Charlie Welch,[3] Steve Parrott,[3] Masuma Pervin Mishu ![ORCID] ,[3] Aziz Sheikh,[7] Romaina Iqbal ![ORCID] [8]

For numbered affiliations see end of article.

**Correspondence to**
Dr Rumana Huque;
rumanah14@yahoo.com

## ABSTRACT

**Introduction** Secondhand smoke (SHS) exposure is a major cause of premature death and disease, especially among children. Children in economically developing countries are particularly affected as smoke-free laws are typically only partially implemented and private homes and cars remain a key source of SHS exposure. Currently, firm conclusions cannot be drawn from the available evidence on the effectiveness of non-legislative interventions designed to protect children from SHS exposure. Following the success of two feasibility studies and a pilot trial, we plan to evaluate a school-based approach to protect children from SHS exposure in Bangladesh and Pakistan—countries with a strong commitment to smoke-free environments but with high levels of SHS exposure in children. We will conduct a two-arm cluster randomised controlled trial in Bangladesh and Pakistan to assess the effectiveness and cost effectiveness of a school-based smoke-free intervention (SFI) in reducing children's exposure to SHS and the frequency and severity of respiratory symptoms.

**Methods and analysis** We plan to recruit 68 randomly selected schools from two cities—Dhaka in Bangladesh and Karachi in Pakistan. From each school, we will recruit approximately 40 students in a year (9–12 years old) with a total of 2720 children. Half of the schools will be randomly allocated to the intervention arm receiving SFI and the other half will receive usual education. Salivary cotinine concentration—a highly sensitive and specific biomarker of SHS exposure—is the primary outcome, which will be measured at month 3 post-randomisation. Secondary outcomes will include frequency and severity of respiratory symptoms, healthcare contacts, school absenteeism, smoking uptake and quality of life. Embedded economic and process evaluations will also be conducted.

**Ethics and dissemination** The trial has received ethics approval from the Research Governance Committee at the University of York. Approvals have also been obtained from Bangladesh Medical Research Council and Pakistan Health Research Council. If SFI is found effective, we will use a variety of channels to share our findings with both

## STRENGTHS AND LIMITATIONS OF THIS STUDY

⇒ This is a definitive trial to assess the effectiveness and cost effectiveness of an innovative school-based intervention designed to influence adult behaviour.
⇒ This is the first multi-country trial of an intervention addressing secondhand smoke exposure in children that will assess a variety of clinical, behavioural and school performance-related outcomes (in children).
⇒ We propose a cluster randomised controlled trial, which requires a large sample size resulting in a greater research burden on the participants than an individual trial.
⇒ The clinical outcomes are only included as secondary outcomes and the trial may not be sufficiently powered to detect a statistically significant difference for all of these.

academic and non-academic audiences. We will work with the education departments in Bangladesh and Pakistan and advocate for including SFI within the curriculum.
**Trial registration number** ISRCTN28878365

## BACKGROUND

Secondhand smoke (SHS) exposure is a serious health hazard to non-smokers, leading to an estimated 890 000 deaths and a loss of 10·9 million disability-adjusted life years (DALYs) globally every year.[1] Children are the worst affected: 28% of deaths from SHS exposure occur in children.[2] SHS exposure impairs children's lung development and causes immune dysregulation, therefore, increasing their risk of acquiring lower respiratory tract infections,[3] tuberculosis, incident cases, recurrent episodes and exacerbations of asthma.[4] Parental smoking is also associated with an increased risk of their children's admissions to hospital.[3] Moreover, SHS exposure in children and adolescents is associated

with poor cognitive functions and reduced academic achievements.[5] Children living in smoking households are at high risk of becoming adult smokers later.[6]

Unfortunately, 40% of children are exposed to SHS worldwide, amounting to a major public health threat.[2] The south and south-east Asia region has the highest burden of disease attributable to SHS in the world. According to Global Tobacco Surveillance System data and Demographic Health Surveys, most children living in Bangladesh and Pakistan are exposed to SHS.[7] In a survey in 12 schools in Dhaka, Bangladesh, we found that 95% of 9–11-year-old children had salivary cotinine levels consistent with recent exposure to SHS.[8] In addition to public places, children are also exposed to SHS in their private homes and cars.

Smoking in indoor public spaces is now banned in many countries, including Bangladesh and Pakistan. Where comprehensive and enforced, these bans have resulted in a significant reduction in SHS exposure and associated morbidity and mortality.[9] However, compliance with the smoke-free legislation is problematic in Bangladesh and Pakistan.[10] Besides, most children are exposed to SHS in their homes and smoke-free laws do not extend to such private spaces. Effective interventions are urgently needed to encourage families to pledge and implement voluntary bans on smoking in homes. However, there is little evidence on the effectiveness of such interventions to protect children from SHS exposure. Two recent reviews found that the evidence remains inconclusive. A Cochrane review included 78 trials (11 from low-income and middle-income countries (LMICs)), many assessing the effect of parental education and counselling programmes.[11] A further systematic review and meta-analysis included 16 trials of interventions delivered by healthcare professionals who provide routine child healthcare. Neither of the two reviews found a significant reduction in children's SHS exposure.[12] Another meta-analysis, which reported on the effect of interventions for reducing SHS exposure at home, found some improvements but recommended further research.[13] Most studies included in the above reviews neither fully described the interventions nor reported on their fidelity. The small number of studies and diversity of intervention approaches precluded definitive conclusions about the most effective interventions and small effects overall. It has been suggested that future studies require context-specific interventions that account for differences in culture, social norms and smoking behaviours across countries.[14 15]

Following the successful execution of the Children Learning About Secondhand Smoke (CLASS II) pilot trial,[16 17] we now aim to conduct a definitive trial. Our overall aim is to make a substantial contribution towards preventing respiratory and other smoking-related illnesses in LMICs by reducing children's exposure to SHS. Our specific objectives are to assess the:

1. Effectiveness and cost effectiveness of a school-based smoke-free intervention (SFI) in:

a. Reducing children's exposure to SHS (primary outcome),
b. Reduction in frequency and severity of respiratory symptoms.
c. The number of contacts with healthcare and improvement in their quality of life.
d. Smoking uptake.
e. School absenteeism and improvement in their school performance.

In addition, we will explore:

2. Implementation of the SFI (fidelity) and barriers and drivers to implementation.
3. Mechanisms of action through which the SFI produces change and contextual factors that influence its implementation and effectiveness.
4. Likely obstacles to and opportunities for implementing and scaling up the SFI, including how best to work with schools and policy-makers to overcome these obstacles and maximise the opportunities.

## METHODS AND ANALYSIS

This is a two-arm cluster randomised controlled trial (cRCT) of SFI with embedded process and economic evaluations. Given that the intervention is delivered by school teachers in classrooms, a cRCT is the most appropriate trial design. The intervention is an educational class-based intervention and, therefore, either school-teachers or children cannot be blinded to the allocation. The primary outcome is mean cotinine levels analysed in a UK-based lab; hence, it will be possible to conceal allocation from the outcome assessors.

### Study sites

The CLASS III trial will be conducted in Dhaka, Bangladesh, and Karachi, Pakistan. These two cities were chosen due to our existing links with the local communities, schools and health facilities.

### Study clusters (schools)

We will recruit 68 schools from the above two sites, 34 from each city. The key eligibility criteria are as follows.

#### Inclusion criteria (schools)

We will include both public and private schools if they:
► Follow national curricula;
► Have year-5 classes for children (approximately 9–12 years old). The average cluster size will be 40 with a minimum of 25 children per cluster.
► Have and abide by smoke-free policies. School teachers involved in the training and in delivering the intervention need to be non-smokers where possible (self-reported).

#### Exclusion criteria (schools)

We will exclude schools if they:
► Have only primary school classes due to the challenges of following up children in other secondary schools.

► Do not use Bangla (in Dhaka) or Urdu (in Karachi) as their education medium

► Have already received training on SFI in a previous project, unless the teachers who were trained have left the school.

Furthermore, it would be desirable to exclude those schools that have year-5 teachers who smoke themselves. However, given the difficulty in verifying smoking status, we will not make this a mandatory exclusion criterion.

## Identifying and recruiting eligible clusters (schools)

We will collect the list of schools situated within purposively selected areas from the Education Ministry and obtain information on their class sizes in year 5, primary or secondary school status, public or private status, boys to girls ratio (B:G) and their medium (language) of teaching. We will prepare a list of eligible schools after initial screening and recruit a random sample of eligible schools. We acknowledge that some of the eligibility criteria can only be assessed after approaching schools and talking to the headmaster and year-5 teachers. We will send a letter addressed to the head teacher of each eligible school, including brief information about the trial and inviting the school to take part. We will meet head teachers face-to-face to provide verbal information and responses to their queries, where needed. We will provide interested schools with a detailed information sheet and consent forms.

Those schools that do not meet the eligibility criteria or those who meet the criteria but do not agree to participate after receiving the trial information will not be enrolled in the trial. Their reasons for not meeting the eligibility criteria or declining to participate will be recorded.

## Withdrawal of clusters (schools)

Once recruited, we will endeavour to keep all schools on board and included in the study. If for any reason, a school withdraws before randomisation, we will recruit a new school to replace the withdrawing school. However, if the withdrawal takes place after randomisation, we will not replace such a school and include their data collected to date in our analyses. Data collected from the schools that withdraw at any point will be included in the analyses.

## Study participants (children)

We will recruit 2720 year-5 (average 40 from each school) school children (age range approximately 9–12 years old) after seeking their parents' consent and their assent through schools. The key eligibility criteria are as follows.

## Inclusion criteria (children)

We will include children if they are:

► Studying in year 5 in the participating school and their age range is approximately between 9 years and 12 years.

► Self-reported non-tobacco users (ie, smoked or smokeless).

## Exclusion criteria (children)

We will exclude children if they have any of the following conditions/situations that the school is aware of:

► Serious medical condition, which is either life-threatening or requires regular hospitalisation.

► History of domestic violence and abuse (in any form).

We will include all consented children in the SFI classroom-based activities. We will, however, exclude children who are active smokers (either self-reported or through a cotinine baseline test) or abuse victims by not including their data within the trial and by not sending any intervention-related materials to their homes.

## Identifying eligible participants (children)

We will request schools to prepare a list of eligible children, including all those who meet the inclusion criteria and excluding those who fall into the exclusion criteria list. Once an eligibility list is prepared, we will give all schools the required number of trial information packs to proceed with recruitment.

## Consenting and enrolling participants (children)

As children participating in this trial will be under 16 years, parental/carer consent is required for them to take part. We will obtain parental consent on an opt-out basis, as follows.

► Participating schools will send out the trial information packs to parents of all eligible children, containing an information sheet and a parent/carer opt-out consent form.

► If parents/carer are not willing for their children to participate in the trial, we will ask them to indicate this by either sending us an opt-out consent form in a return-addressed envelope or call/text/email us on the contact details provided within the information pack.

► We will give parents/carers a minimum period of 7 days to indicate if they do not wish their children to take part in the study before sending them a reminder.

The children's assent form will be administered within school at the same time when the trial information packs are sent to the parents/carers. If children are unwilling, they will be able to either let their teachers or parents/carers know, as they feel appropriate. If parents/carers indicate their disapproval for their child to take part in the study, this will supersede the child's assent to participation. Any child who or whose parents/carers have declined to participate will be removed from the list of eligible children by the school and the final list will be handed over to the research team.

Recruitment will be staggered, and each country will have a recruitment target of at least 4 schools and 200 children per month. Two weeks prior to the recruitment week, our team will liaise with the respective school in preparing lists of eligible children and sending them and their parents/carers the trial information packs.

### Ineligible and non-consenting participants (children)

Those children who do not meet the eligibility criteria or those who meet the criteria, but either they or their parents/carers do not agree to participate after receiving the trial information, will not be enrolled in the trial. Their reasons for not meeting the eligibility criteria or declining to participate will be recorded. This information will be kept anonymous.

### Withdrawal of participants (children)

A child can be withdrawn from participation at any time even after enrolment or allocation. If a child is withdrawn from the intervention for any reason, their follow-up assessments and data collection will continue as per protocol unless parents/carers or children specifically ask for their withdrawal from the study completely. We will explain the objectives and outcome of the study to the teachers and students clearly, and engage with them at regular intervals to ensure participant retention and complete follow-ups. However, if the child is withdrawn completely from the study, then no more data will be collected. Their data will still be included in the analysis and counted as lost to follow-up.

### Cluster (schools) randomisation and allocation

Once baseline data are collected, participating schools will be randomly allocated (1:1) by the York-based trial statistician to the 2 trial arms (approximately 34 in each arm) using minimisation (with a random element incorporated to help maintain allocation concealment). The minimisation will be used to balance treatment allocation on the country (Bangladesh or Pakistan), school type (public or private), B:G in year 5 (B:G<0.95, $0.95 \leq$ B:G$\leq 1.05$, B:G>1.05) and the number (N) of students in year 5 in the participating school cluster (N<30, $30 \leq$ N$\leq 60$, N>60). Because of the nature of the intervention, it will not be possible to mask the children and schoolteachers from the allocated intervention. To avoid bias, we will ensure that all baseline data are collected before treatment allocation.

### Intervention details

Once children are enrolled, schools will be randomised to receive either the SFI or treatment as usual. These treatment conditions are described as follows.

### Smoke-free intervention

All participating children in the intervention arm will receive the SFI delivered by their teachers. Teachers will receive prior training in delivering the intervention. This training will focus on addressing their knowledge gaps around tobacco, improving skills in using a variety of teaching methods and their ability to build confidence within and teach negotiation skills to children.

The intervention will consist of:

▶ Two 45-minute sessions delivered over 2 days by schoolteachers. The duration of these sessions has been planned to fit within a regular school lesson. These sessions will consist of a flip chart presentation and a drama activity, respectively. The session activities are specially designed to increase pupils' knowledge about SHS and related harms and to negotiate smoke-free homes with their parents/carers who will implement smoking restriction rules. The seven acts of the drama, to which parents/carers are invited to attend, will give children the opportunity to practise their negotiating skills and be confident within their cultural context. It will also serve as a visual incentive for the parents/carers not to smoke inside homes.

▶ Children will be given an achievement certificate to record the seven achievements to make their homes smoke free. Children will also receive a promise form that describes the main step to achieve a smoke-free home, that is, to walk seven steps away from the house door to smoke. It will also contain a tear-off slip to make a commitment to impose smoking restrictions at home. Children will take promise forms to their parents/carers, show them the messages and negotiate with them to 'sign-up' to the smoke-free homes 'promise' form.

▶ A set of 4 follow-up sessions (15 min each) to reinforce key messages delivered in the initial sessions to be delivered once a week over 6–7 weeks after the 2 initial sessions. The immediate first follow-up session will be based on feedback from parents/carers about the drama activity. The second and third follow-up sessions will consist of a word search game and a quiz game in which children will be asked questions and given answer options. The final follow-up session will be based on small group discussions among the students about their experiences of negotiating smoke-free homes and if they faced any challenges.

### Usual education

Schools in the control arm will receive the SFI at the completion of the trial.

### Outcomes assessments

A causal link between SHS exposure and respiratory infections is well established.[3] In the form of salivary cotinine, we also have a highly sensitive and specific biomarker of SHS exposure.[18] Therefore, we propose SHS exposure (proximal) and respiratory symptoms (distal) as the primary and secondary outcomes, respectively. The outcomes for this definitive trial will be measured before and after the intervention in each of the study's arms (see table 1 for the schedule of assessments).

### Primary outcome

We intend to use children's salivary cotinine as a biomarker of SHS exposure at baseline and month 3. Cotinine—the major proximate metabolite of nicotine—has a relatively long half-life (17 hours), which allows detection of tobacco exposure even after 3 days.[18] Once children are enrolled in the study, saliva samples will be obtained from all participating children at baseline and again at 3 months post-randomisation. Samples can be stored at

**Table 1** Schedule of assessments within the CLASS III trial

| Assessments | Baseline | Post-intervention | | |
| --- | --- | --- | --- | --- |
| | | Month 3 | Month 6 | Month 12 |
| Eligibility and consent | X | | | |
| Sociodemographic and medical history | | | | |
| ► Personal details | X | | | |
| ► Household details | X | | | |
| ► Medical conditions and history of medications | X | | | |
| Smoking-related behaviours | | | | |
| ► Smoking restrictions and social visibility | X | X | X | X |
| ► Attitudes toward smoking | X | X | X | X |
| ► Health service use | X | X | X | X |
| ► Quality of life | X | X | X | X |
| ► Exposure to SHS—salivary cotinine | X | X | | |
| ► Respiratory symptoms diary | | X | X | X |
| Academic assessment | | | | |
| ► APQ | X | X | X | X |
| ► School absenteeism report | X | X | X | X |
| AE reporting | | X | X | X |
| Process evaluation | | X | X | |

AE, adverse event; APQ, Academic Performance Questionnaire; CLASS III, Children Learning About Secondhand Smoke; SHS, secondhand smoke.

ambient temperature for a period of 2 weeks before transported to a specialist laboratory—ABS Labs (https://www.abslabs.co.uk/) in the UK—these are to be analysed using a gas–liquid chromatography technique. Samples will not contain any participant-identifiable information and will only have the trial enrolment number.

## Secondary outcomes

There will be a number of secondary outcomes, which are the same as we collected in the CLASS II pilot trial,[5] including (a) the frequency and severity of respiratory symptoms, (b) self-reported smoking restrictions and social visibility at home, (c) health service use, (d) quality of life and (e) academic performance and school absenteeism. These will be measured at 3 months, 6 months and 12 months (see table 1).

## Data collection methods

Prior to randomisation, a baseline assessment will include a classroom-administered questionnaire (including EQ-5D-Y,[9] health service use and smoking behaviour) to be completed by participating children, Academic Performance Questionnaire (APQ) and school absenteeism form completed by schoolteachers and saliva sample collection by the research team for each child. Each child will also receive a respiratory symptoms diary with instructions on how to use it. Follow-up assessment will take place at 3 months, 6 months and 12 months post-randomisation from all eligible and consenting participants. All assessments carried out at the baseline will be repeated at the

follow-up assessment except cotinine levels, which will only be assessed at month 3 post-randomisation.

## Statistical considerations
### Sample size

Informed by the results of the CLASS II trial,[16] we assume an average cluster size of 40, that 5% of children within a given cluster are not eligible (ie, they have a salivary cotinine concentration less than 0.05 ng/mL or greater than 12 ng/mL, report tobacco use or have a history of domestic abuse) and that 10% of children who are eligible (within a cluster) do not provide a salivary cotinine reading at 3 months post-randomisation. Under these assumptions, we would expect to obtain (on average) primary outcome data for 34.2 participants per cluster. Rounding this figure up to 35, and assuming a coefficient of variation in cluster size of 0.4 and intracluster correlation of 0.05, gives a design effect of 2.98. Assuming the marginal variance of the primary outcome is $1.38^2$, a total of 766 participants would be required for an individually randomised trial to obtain 80% power to detect a difference in salivary cotinine concentration of 0.28 ng/mL in a 2-sided t-test (against of size 5%). Hence, approximately 2284 primary endpoints (ie, valid salivary cotinine measurements at 3 months post-randomisation) are required to obtain 80% power for the cluster randomised design. Assuming 34 observations per cluster, approximately 2284/34 = 67.2 clusters are required, hence, the total recruitment target of 68 clusters (2720 children).

## Statistical analysis

The outcome data will be analysed once at the end of the trial follow-up period (ie, no interim analyses of accumulating outcome data will be undertaken), with participants being analysed as part of the group to which they were assigned. We will compare the average cotinine levels between the groups using a linear multilevel model controlling for pertinent baseline covariates (at the child and school level), minimisation factors and adjusting for clustering by schools. The distributional assumptions will be checked and different link functions (eg, log link) will be used if indicated. Checks for data quality and completion rates will be carried out on a regular basis.

## Economic evaluation

We will undertake a full cost-effectiveness analysis using methods that have already been piloted.[5] The first stage estimates the training cost and the cost of delivering the intervention. We will use a service-use questionnaire developed from the questionnaire used in the pilot to record the utilisation of healthcare resources. The self-administered questionnaire will be completed by students. We, therefore, will not ask for details regarding, or limit our questions to, specific diagnoses or settings. We will also record and calculate the costs of medications related to these illnesses, which are dispensed. Quantities of resource use (contacts) are multiplied by local unit costs to derive an individual cost profile for each participant.

We will measure health-related quality of life in children by using the EQ-5D-Y,[9] which will be administered at baseline and each follow-up period. The results will be used to calculate quality-adjusted life years (QALYs) for children in the trial, using the area under the curve. In addition, the symptoms for lower respiratory infection and otitis media collected in the symptom diary will be used to estimate DALY changes for all children in the trial.

We will conduct an incremental cost-effectiveness analysis of SFI over and above the control. Costs and QALYs will be combined to calculate the incremental cost-effectiveness ratio in terms of incremental cost per QALY gained.

## Process evaluation

A mixed-methods process evaluation will explore three key functions: mechanisms of action, context and implementation.[19]

Mechanisms of action (mediators and acceptability): all children (in the intervention and control arms) will complete a short questionnaire at the 3-month and 6-month follow-ups, measuring mechanisms of action constructs as mediators of behaviour change (see table 2). These are based on the evidence for the links between the behaviour change techniques[20] that constitute the active ingredients of the intervention, and the respective mechanisms of actions[21] of those techniques (see table 2 and figure 1).

In terms of the mediators, or knowledge,[19] we will adapt the Smoking Attitudes Knowledge and Practice Instrument (knowledge component).[20] For beliefs about consequences, we will adapt the Smoking-Related Health-Beliefs Scale.[22] For skills construct, we will adapt a behaviour specific Self-efficacy Scale.[23] For intentions, we will adapt a behaviour-specific intentions scale.[24] For self-regulation, we will adapt the Goal-Setting and Planning and Scheduling Scale.[25] For the behavioural cueing construct, we will adapt the Self-Report Habit Index.[26]

In the 3-month questionnaire, children in the intervention arm will report which SFI activities they have engaged with, the acceptability of those activities and the perceived impact on smoking in their family home.

At the 3-month follow-up, we will conduct a focus group discussion (FGD) with 6–8 children in 6 Bangladesh and 6 Pakistan intervention arm schools (purposively selected to reflect a mix of private/public, B:G and selected/not selected for fidelity checks). The FGDs will explore key issues that emerge from the acceptability questionnaire, for example, the children's experiences of participating in the SFI and negotiating a smoke-free home (SFH) with their family. We will also conduct an FGD with parents in these schools to explore their views and experiences of their children participating in the SFI, their conversations about creating an SFH and any changes that may have occurred.

Context: contextual factors (eg, socioeconomic status) will be measured for all children in the baseline, 3-month and 6-month follow-up questionnaire. In the 12 selected schools, the teachers and head teachers (2 and 1 per school respectively) will be interviewed once the SFI has been delivered. These interviews will explore how contextual factors such as the school environment and other social, economic, cultural, environmental and political factors have influenced the delivery and impact of the SFI. The FGDs with children and parents will discuss contextual influences on the impact of the SFI

Implementation (feasibility, fidelity): the teacher/headteacher interviews will also explore implementation issues as well as perceptions of the potential reach of SFI, likely obstacles and potential opportunities for scale-up.

Fidelity to delivering the six SFI sessions will be assessed using a fidelity index, linked to the behaviour change techniques[27] that underpin the SFI. A third of intervention schools (six in each country) will be randomly selected and their six SFI sessions will be observed by two independent checkers using the fidelity index.

We will also interview 2–3 policy-makers in each country to explore their views on opportunities for implementing and scaling up the SFI and how best to work with schools and policy-makers to overcome the obstacles and maximise the opportunities.

All FGDs and interviews will be conducted face-to-face using a topic guide to ensure consistency, with flexibility to allow participants to generate naturalistic data on what they see as important. Prior to commencing data collection with teachers and parents, we will collect consent (this will have been previously secured for headteachers and children). In these discussions, we will use a

**Table 2** The logic model of the SFI

| Resources | Activities | Outputs (for process) | Short term outcomes | Medium term outcomes | Long term outcomes | Impact |
|---|---|---|---|---|---|---|
| Year-5 schoolteachers's relevant resource materials; teacher training; teachers training to pick up any signs of distress among children as an untoward consequence of SFI; 2×45 m sessions over 2 consecutive days | Storytelling, drama and role-play activities focused on building children's confidence in raising their concerns about SHS with their parents/carers and enhancing their negotiation skills, and allowed children to learn and practise relevant negotiating strategies. 1. Information about health consequences 2. Salience of consequences 3. Behavioural practice/rehearsal 4. Goal setting (behaviour) | Evidence of practising strategies, developing skills and confidence around SHS negotiation; evidence of knowledge of harms | Smoke-free homes (SFH)/SHS negotiation self-efficacy; SFH/SHS negotiation intentions; evidence of knowledge of harms; knowledge; Mechanisms of Actions (MOA); intentions; skills; beliefs about consequences; knowledge | Self-reported smoking restrictions; salivary cotinine; AE monitoring: distress arising from smoking-related illness (SRI) | Frequency severity of respiratory symptoms; lung function tests | APQ; school absenteeism; quality of life (EQ-5D-Y); health service use |
| Four refresher sessions (15 min each) over the subsequent 4 weeks | The discussion, quiz and games aimed to make children aware of the harms of SHS and motivate them to achieve a smoke-free home. Revising the salient points of the initial sessions, encouraging children to share their experience of initiating relevant conversations within their families encouraging children to share their experience of initiating relevant conversations within their families. 1. Information about health consequences 2. Salience of consequences 3. Behavioural practice/rehearsal 4. Goal setting (behaviour) | Evidence of knowledge of harms of SHS from quiz answers; evidence of motivation to achieve SFH from games; evidence of sharing relevant experiences of initiating conversations around SHS and SFH | SHS risk awareness, SHS negative outcome expectancies; SFH intentions; SFH/SHS negotiation self-efficacy; SFH/SHS negotiation intentions; MOA; knowledge; beliefs about consequences; intentions | | | |
| Home promise forms for families (described in activities) | Reading of graphic representations of the hazards of SHS, pictorial guidance to help them make their homes smoke free and a tear-off slip to commit to imposing smoking restrictions at home visitors and cars. 1. Goal setting (behaviour) 2. Problem solving 3. Action planning 4. Behavioural contract | Evidence that the home promise form was taken home | Action planning to negotiate SFH/SFH self-efficacy; SFH intention; MOA; intentions; beliefs about consequences; behavioural regulation; behavioural cueing | | | |

APQ, Academic Performance Questionnaire; SFI, smoke-free intervention; SHS, secondhand smoke; SRI, smoking-related illness.

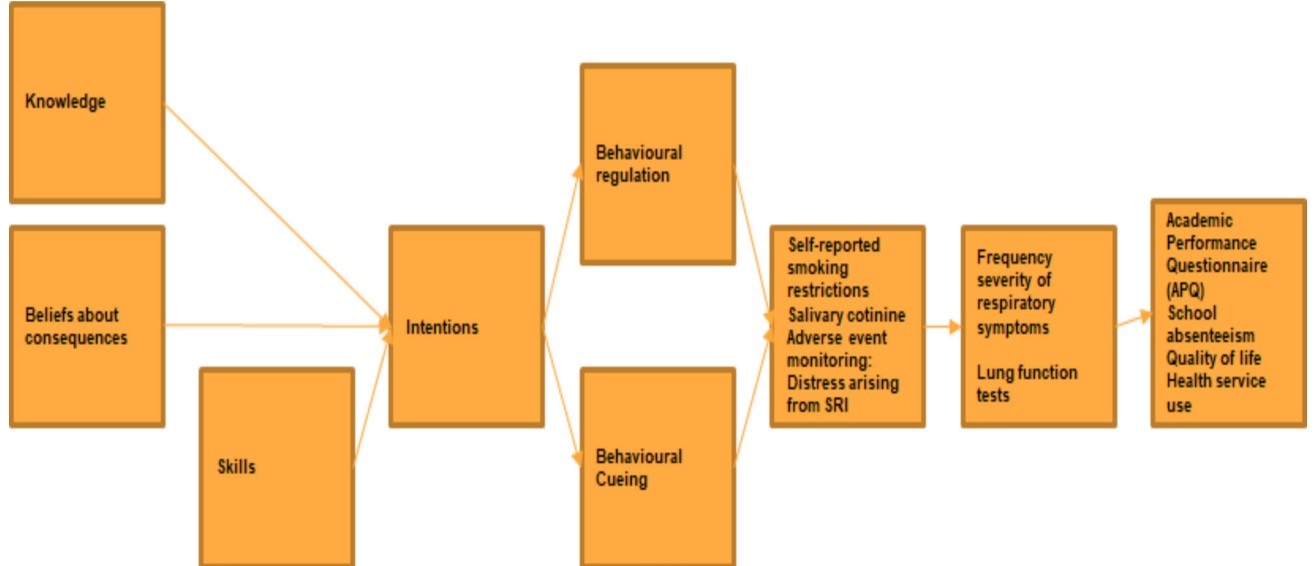

**Figure 1** Causal mechanisms for the intervention. SRI, Smoking-related illness. APQ, Academic Performance Questionnaire.

hermeneutics approach,[28] which encourages participants to discuss features of the intervention to elicit data on their experience of its delivery/receipt. With the participants' permission, the FGDs and interviews will be audio-recorded digitally and transcribed verbatim.

The quantitative acceptability questionnaire and fidelity data will be analysed descriptively. Mediator analysis will test the proposed causal model. Verbatim transcripts, of FGD and interview data, translated into English, will be analysed using the framework approach, whereby data are organised into matrices (for headteachers, class teachers, students and parents). Integration of the datasets will be done using a 'triangulation protocol'.[29]

## Data management

The field data collectors will collect trial data (including quantitative process evaluation data) using mobile digital tablets. An online data tracker will help in monitoring timely data capture on all recruited participants. These processes will be trialled before the commencement of the trial. All data will be transferred to the trial statistician at the University of York who will conduct the analysis using the statistical software packages: STATA and R.

Qualitative process evaluation data will be collected using digital recorders and transferred to the secure (password-protected) computer of the local qualitative researcher, to be analysed in the Framework matrices set up in Excel 2021. The recording will be deleted from the recorder once the verbatim transcript is produced. Researchers will complete the fidelity index in Excel V.2021.

## Data quality and standards

For saliva sample quality measurements, samples will be stored for a maximum period of 2 weeks before being transported to the specialist laboratory in the UK. Overall data quality will be ensured through training and supervision. Data entry validation will occur by double-checking

a random sample of the data in the field. Moreover, quantitative data once entered will also be checked by the statistician.

The quality of the qualitative data will be ensured by training the researchers in Bangladesh and Pakistan and through detailed feedback from a senior researcher at the University of York to ensure good questioning techniques. In all steps of the data analysis, rigorous procedures to ensure 'trustworthiness' of the findings will be undertaken—the framework matrices (for each participant group) will be produced as a team, a 10% subsample of data charting into the matrices will be checked and a sample of sections of the write-up will be jointly produced (with the senior researcher in York). Completed framework matrices will be shared and discussed across the partner organisations to ensure credibility.

## Data security and confidentiality of potentially disclosive information

Data management will comply with the Data Protection Act (2018) and the General Data Protection Regulations. Each participant will be asked to understand and sign an approved consent form before they take part in the CLASS III trial. Parental consent will be sought prior to children taking part. Data collected using questionnaires and interviews/FGDs will be pseudo-anonymised to remove information, which could identify the participant. Consent forms, digital recorders and transcripts will be kept inside a locked cabinet in the relevant office. Electronic data will be collected on password-protected devices. Data will be transferred securely to the database, which will be installed on a secure server (password-protected). Digital audio recordings will only be listened to by members of the research team. All documents and audio recordings will be retained for a minimum of 5 years and then destroyed, according to the University of

York's policy. Saliva samples will be preserved in the UK laboratory for 3 months post-analysis.

## AEs procedures

We are expecting a minimal number of adverse events (AEs) and no serious adverse events (SAEs) during the study. SFI is an educational intervention and has been very well received in our previous studies[16 30] without leading to any directly related AEs. Nevertheless, there will be a vigilant surveillance system in place for AEs occurring during the course of the trial with particular emphasis on identifying, recording, reporting and managing any SAEs. We will sensitise school teachers to look for signs of any AEs resulting from the interactions between children and their parents/carers. We will also encourage children and parents/carers to report any related adversities.

### Detecting, recording and reporting of AEs and SAEs

In the event of any AE reported by the child, their parents/carers, schoolteachers, and research assistants will complete an AE form, which will include a medical diagnosis, if relevant and available. The research assistant will also call the trial manager on the same day providing a verbal report of the event. The trial manager will ensure that the event is classified appropriately after receiving the verbal report. All AEs will be reported to the principal investigators (Bangladesh and Pakistan) within 3 days of detection. AE data will be collated and reported to the trial sponsors and National Bioethics Committee at 6 monthly intervals, and also be reported to the Study Operational Committee and the Independent Trial Steering Committee (ITSC) at their regular meeting. All SAEs must be reported to the principal investigator within 24 hours of detection and should also be reported to the trial sponsors and National Bioethics Committee within three working days. All serious events must also be reported to all study investigators and the chair of the ITSC within 3 days. The Chief Investigator will have the overall responsibility to ensure that all AEs are reported according to the above protocol.

## Patient and public involvement

'No patient was involved'.

## Study organisational structures

Our trial management relies on (a) a trial coordination team (York) consisting of a trial coordinator, methodologist, statistician, qualitative researcher and an economist; (b) York Trials Unit providing methodological, statistical and data management support and (c) trial and data management teams (Bangladesh and Pakistan) consisting of trial manager(s), research assistants, field data collectors and data entry operators.

An eight-member ITSC has been formed involving subject experts, a statistician and a health economist. The ITSC is expected to oversee the trial, assess the progress of the trial against the agreed timeline, adherence to the trial protocol, and ensure patient safety and ethical considerations. The TSC will also guide the study team to solve any emerging issues.

## Protocol Amendment

All amendments to the protocol will be first discussed with the chief investigator and then submitted to the ethics committee for formal approval. A judgement will be made on the nature of the amendment, that is, major or minor as per guidance from the ethics committee. All minor amendments will be implemented once notified to the ethics committee and all major amendments will be implemented once approved by the ethics committee.

## Protocol violations and deviations

The research team will not deviate from the protocol without agreement with the chief investigator and securing an agreement with the ethics committee and Study Operational Committee except in situations where a deviation is necessary to remove an immediate hazard to the participants. Any such deviations (both nature and reason) would be recorded in the AE form and if necessary an amendment to the protocol will be secured through a formal process.

## Quality assurance and ethics

The study will be conducted in accordance with current Medical Research Council (MRC) Good Clinical Practice guidelines and the NHS Research Governance Framework. Administrative approval will be sought from each participating school. The study will be subject to all research management and governance procedures in place at the University of York, including the requirement for audit.

The trial will be conducted to protect the human rights and dignity of the participants as reflected in the 1996 version of the Declaration of Helsinki. Participants will not receive any financial inducement to participate in the trial. In order to protect the trial participants, the following provisions will be made/upheld: the trial has been designed to minimise the burden of participants and any foreseeable risk in relation to the intervention involved; the explicit wishes of the participant will be respected, including the right to withdraw from the trial at any time; the interest of the participant will prevail over those of science and society and provision will be made for indemnity by the investigator and sponsor.

Ethical approvals have been received from the University of York and Bangladesh MRC and Pakistan Health Research Council.

## DISSEMINATION

The issue of SHS is already a national priority in Bangladesh and Pakistan. If SFI is found to be effective, we will use advocacy, our existing linkages and impact enhancement schemes to maximise the impact of our results in these countries and beyond. We have partnered with non-governmental organisations (NGOs) within the two

countries with expertise in advocating for tobacco control measures. Together, we will develop a dissemination strategy, which will target academic and non-academic audiences using a variety of media. Beyond the two countries, our team is also connected to international funders and agencies supporting tobacco control efforts. We will use these networks and existing partnerships to disseminate and seek support for our research findings.

**Author affiliations**
[1]Department of Economics, University of Dhaka, Dhaka, Bangladesh
[2]ARK Foundation, Dhaka, Bangladesh
[3]Department of Health Sciences, University of York, York, UK
[4]Hull York Medical School, University of York, York, UK
[5]Valid Research, Wetherby, UK
[6]School of Psychology, University of Leeds, Leeds, UK
[7]Division of Community Health Sciences, University of Edinburgh, Edinburgh, UK
[8]Department of Community Health Sciences and Medicine, Aga Khan University, Karachi, Pakistan

**Contributors** KS, RH and RI conceived the study and wrote the first and all subsequent drafts. MKhokhar, CJ, MKanaan, CH, IK, CW, SP, MPPM and AS made comments and suggestions on the draft paper. All authors participated in manuscript revisions, edits and read and approved the final manuscript.

**Funding** The study is funded by a grant received from the Medical Research Council (MR/T004959/1). The study is sponsored by the University of York, Heslington Hall, York, UK.

**Competing interests** None declared.

**Patient and public involvement** Patients and/or the public were not involved in the design, or conduct, or reporting, or dissemination plans of this research.

**Patient consent for publication** Not applicable.

**Provenance and peer review** Not commissioned; externally peer reviewed.

**Data availability statement** Data are available upon reasonable request.

**ORCID iDs**
Rumana Huque http://orcid.org/0000-0002-7616-9596
Kamran Siddiqi http://orcid.org/0000-0003-1529-7778
Masuma Pervin Mishu http://orcid.org/0000-0002-6545-9117
Romaina Iqbal http://orcid.org/0000-0002-5364-4366

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
