## [Reviewer comments · BMJ Open]

ARTICLE DETAILS

TITLE (PROVISIONAL)	Children Learning About Second-hand Smoke (CLASS III): Protocol for a cluster randomised controlled trial of a school-based Smoke-Free Intervention in Bangladesh and Pakistan
AUTHORS	Huque, Rumana; Siddiqi, Kamran; Khokhar, Mariam; Jackson, Cath; Kanaan, Mona; Hewitt, Catherine; Keller, Ian; Welch, Charlie; Parrott, Steve; Mishu, Masuma Pervin; Sheikh, Aziz; Iqbal, Romaina

VERSION 1 – REVIEW

REVIEWER	Kaung Lwin The University of Tokyo, Global Health Policy
REVIEW RETURNED	30-Oct-2022

GENERAL COMMENTS	This protocol is well-written with clearly defined objectives. I think it is ready for publication.
---

REVIEWER	Ömer Alkan Ataturk University, Department of Econometrics
REVIEW RETURNED	10-Feb-2023

GENERAL COMMENTS	Thanks for the opportunity to review the manuscript “Children Learning About Second-hand Smoke (CLASS III): Protocol for a cluster randomised controlled trial of a school-based Smoke-Free Intervention in Bangladesh and Pakistan” submitted to the BMJ Open. I have carefully reviewed this manuscript and below is my decision. - The article title should be changed.- In the abstract, remarkable findings can be given in detail.- The hypothesis of the research is not clear. The author(s) needs to explain what are the main research questions of this paper.-The original value of the study and its contribution to the literature should be explained in more detail.-The introduction section should be rewritten.-I would suggest adding to the literature and referencing it within the discussion as well. There is study that have examined sexual violence. * “Secondhand smoke exposure for different education levels: findings from a large, nationally representative survey in Turkey”. BMJ open * “Tobacco smoke exposure among women in Turkey and determinants”. Journal of Substance Use. It can be published after corrections are made.
--

REVIEWER	Marie Chan Sun University of Mauritius, Department of Medicine
REVIEW RETURNED	17-Feb-2023

GENERAL COMMENTS	- This research work is commendable as it is well-written and well-designed. - The overall aim which is "to prevent respiratory and other smoking-related illnesses (SRI) in LMICs by reducing children's exposure to SHS" seems over-ambitious. Authors will have to formulate a more realistic aim. - There is a need for the authors to justify the sample size for the statistical significance of the expected findings. - The choice of the two cities was explained but the number of schools and children participating in the trial was not explained. It was only mentioned that this work was informed by the CLASS II trial. - There is a need for the investigators to explain to the children in simple terms the trial objectives and process [Page 18 last paragraph]. - Please clarify the use of Excel for qualitative data analysis [Page 6 Paragraph 3]. Thank you M Chan Sun
--

VERSION 1 – AUTHOR RESPONSE

Reviewer: 1	
This protocol is well-written with clearly defined objectives. I think it is ready for publication.	We thank the reviewer for reviewing the paper and finding it ready for publication.
Reviewer: 2	
(1) The article title should be changed	We are carrying out this definitive trial after successful completion of the feasibility and pilot trials. To be consistent with the previous studies and publications, the study team prefers to stick to the current title.
(1) In the abstract, remarkable findings can be given in detail.	This is a protocol paper. We will report the findings in the main trial findings paper.
(2) The hypothesis of the research is not clear. The author(s) needs to explain what are the main research questions of this paper.	The overall aim and objectives of the study are stated in page 7. Our objectives can be rewritten as research questions. However, to remain consistent with our funding application and the trial registry, we choose to retain the objectives.
(3) The original value of the study and its contribution to the literature should be explained in more detail.	The strengths of the paper are stated on page 5. On page 6, we have added further explanation to highlight the

	original value of the study.
(4) The introduction section should be rewritten	We thank the reviewer for this comment. We have added some more text to justify our study. However, on the need to rewrite the introduction, we would appreciate more clarity from the reviewer.
(5) I would suggest adding to the literature and referencing it within the discussion as well. There is study that have examined sexual violence:  • Secondhand smoke exposure for different education levels: findings from a large, nationally representative survey in Turkey". BMJ open • "Tobacco smoke exposure among women in Turkey and determinants". Journal of Substance Use. 	Thank you for the suggestion. We have added one reference (15) as suggested. As sexual violence is not the focus of the study, we have not included the second reference as suggested by the reviewer.
Reviewer: 3	
(1) This research work is commendable as it is well-written and well-designed. The overall aim which is "to prevent respiratory and other smoking-related illnesses (SRI) in LMICs by reducing children's exposure to SHS" seems over-ambitious. Authors will have to formulate a more realistic aim.	We acknowledge that this overall aim of the study will not be achievable during the study period. We believe that this research can still make an important contribution towards reducing SHS exposure in the two countries. We have modified the text accordingly.
(2) There is a need for the authors to justify the sample size for the statistical significance of the expected findings	We have explained the sample size on page 15 and 16.
(3) The choice of the two cities was explained but the number of schools and children participating in the trial was not explained. It was only mentioned that this work was informed by the CLASS II trial.	Number of schools (68 in total) and children (2,720 in total) are explained on page 16 under the sub-heading 'sample size'.
(4) There is a need for the investigators to explain to the children in simple terms the trial objectives and process [Page 18 last paragraph].	We thank the reviewer for the comment. The information sheet explains the study objectives in simple terms to make those understandable to the children.
(5) Please clarify the use of Excel for qualitative data analysis [Page 6 Paragraph 3].	We have clarified that the Framework matrices for the qualitative analysis will be set up in Excel (page 21) (1 per participant group).

VERSION 2 – REVIEW

REVIEWER	Marie Chan Sun University of Mauritius, Department of Medicine
REVIEW RETURNED	16-Apr-2023
GENERAL COMMENTS	I am of the view that Serious Adverse Events need to be reported within 24 hours.